# Position: AI Safety Should Prioritize the Future of Work

**Sanchaita Hazra** [1]   **Bodhisattwa Prasad Majumder** [2]   **Tuhin Chakrabarty** [3]

## Abstract

Current efforts in AI safety prioritize filtering harmful content, preventing manipulation of human behavior, and eliminating existential risks in cybersecurity or biosecurity. While pressing, this narrow focus overlooks critical human-centric considerations that shape the long-term trajectory of a society. In this position paper, we identify the risks of overlooking the impact of AI on the future of work and recommend comprehensive transition support towards the evolution of meaningful labor with human agency. Through the lens of economic theories, we highlight the intertemporal impacts of AI on human livelihood and the structural changes in labor markets that exacerbate income inequality. Additionally, the closed-source approach of major stakeholders in AI development resembles rent-seeking behavior through exploiting resources, breeding mediocrity in creative labor, and monopolizing innovation. To address this, we argue in favor of a robust international copyright anatomy supported by implementing collective licensing that ensures fair compensation mechanisms for using data to train AI models. We strongly recommend a pro-worker framework of global AI governance to enhance shared prosperity and economic justice while reducing technical debt.

## 1. Introduction

Machine Learning (ML) researchers working on AI safety primarily focus on misuse or existential risks posed by advanced AI models. By aligning AI with humans, the community of researchers aims to ensure AI systems are moral, beneficial, and reliable. Significant research is conducted to investigate how AI systems can be vulnerable to adversarial attacks that bypass their ethical guidelines and perform restricted actions (Anil et al., 2024; Wei et al., 2024; Qi et al., 2023) or how powerful AI systems can manipulate and persuade humans towards undesirable outcomes (Salvi et al., 2024; Palmer & Spirling, 2023). There has also been an emphasis on evaluating the risks of AI against bioterrorism (Peppin et al., 2024), automated warfare, and possible rogue AIs (Hendrycks et al., 2023). Researchers have also studied how large language models (LLMs) learn to mislead humans via RLHF (Wen et al., 2024) or how they *fake* alignment, i.e., generate desirable outputs while being trained, only to later produce non-compliant outputs when not under observation (Greenblatt et al., 2024). However, relatively less emphasis has been placed on the medium and long-term impacts of generative AI on society. In this position paper, we argue **AI safety should prioritize the future of work** and recommend a systemic overhaul of AI research practices as well as governance to protect meaningful labor.

The rapid proliferation of generative AI systems that create images, text, and code at scale has led many to prematurely anthropomorphize these tools and attribute consciousness or sentience to them. The allure of such behavior stems from a psychological tendency to seek agency in complex systems, while the widespread adoption reflects basic economic principles of substitution, where firms naturally gravitate toward technologies that can automate skilled and cognitive labor at dramatically lower marginal costs. LLMs, like ChatGPT, are transforming online labor markets by introducing automation capabilities that potentially impact traditional freelance work roles. Recent work from Demirci et al. (2024) shows that the introduction of ChatGPT led to a 21% decrease in the number of writing and coding job posts, while the introduction of Image-generating AI technologies led to a significant 17% decrease in the number of image creation jobs. AI tools are being adopted by creative industries, with companies in gaming, movies, interior design, and advertising beginning to use them in place of human artists (Edwards, 2023; Roose, 2022; Tobin, 2023).

Innovation is deemed a socially beneficial idea, where the self-serving efforts of the private sector may result in increased social benefit when directed through well-functioning market channels. The private advantage of innovators lies in being a monopolist, which requires special skills and copious resources. Originated by Gordon Tullock in 1967 (Tullock, 2008) and made famous by Anne

---

[1]University of Utah [2]Allen Institute for AI [3]Stony Brook University & Salesforce AI Research. Correspondence to: Sanchaita Hazra <sanchaita.hazra@utah.edu>.

*Proceedings of the 42nd International Conference on Machine Learning*, Vancouver, Canada. PMLR 267, 2025. Copyright 2025 by the author(s).

Krueger in 1974 (Krueger, 2008), rent-seeking is a simple but powerful idea where individuals, businesses, or groups gain financial benefits through political or legal manipulation involving lobbying, regulation, or various government interventions at the expense of broader social welfare, economic growth, and inefficient allocation of resources. In the wake of recent AI developments, governments and interest groups aim to seek economic advantages through political influence and regulation through the lens of innovation. Leading firms lobby for favorable policies (Wiggers, 2025), training data monopolization, and expensive AI patents. By favoring certain companies with preferential subsidies and national security contracts, governments also encourage rent-seeking.

AI researchers should care about labor market impacts because they directly connect to the broader mission of creating beneficial and aligned AI systems. AI safety cannot be divorced from labor market dynamics and economic justice. While traditional AI safety focuses on preventing harmful outputs or existential risks, through this position paper, we argue that the greatest immediate risk may be the systematic disruption of human agency and economic dignity in the workforce. Replacing human labor with AI could weaken both direct human control (through voting and consumer choices) and indirect influence that comes from human participation in societal systems (Kulveit et al., 2025).

Towards this, we first discuss **systematic risks** that generative AI causes to future work. These include (1) increasing technical debt, (2) the speed of AI automation outpacing society's ability to adapt, (3) uneven democratization of AI resources, (4) generative AI as extractive institutions leading to a decline in shared prosperity, (5) generative AI impairing learning and knowledge creation, and (6) erosion of creative labor markets through the exploitation of copyrighted works. We then provide **concrete recommendations** that require collective actions: (1) worker support in response to job displacement, (2) promotion of worker interests, (3) open-source training data and a fair royalty-based compensation system to safeguard creative labor, (4) increased support on improving detection and watermarking AI-generated content, and (5) broad stakeholder engagement in AI policy-making to avoid regulatory capture.

## 2. Risks

In this section, we outline risks stemming from generative AI and how that impacts the future of work. These discussions are powered by socio-economic theories and will lead to the recommendations we made in the next section.

### 2.1. Increasing Techincal Debt

In daily conversations, debt stipulates a situation of owing money, especially when one cannot afford to pay out of one's pocket. Vee (2024) abstracts it to "borrowing against the future". Menshawy et al. (2024); Li et al. (2024), and others represent technical debt as long-term consequences of choosing quick, short-term solutions in software development instead of implementing more robust and scalable approaches. In the current era of LLMs, the arms race between the leading developers and stakeholders contributes to the increasing technical debt. Rushed deployments of models, lack of testing, vague and opaque claims on data transparency and its reliability, unchecked model hallucinations, and misuse in sensitive applications such as medical, law, and financial decision-making - all contribute to the creation and magnifying of moral hazard.

Vee (2024) rightly argues that models, once released on the internet, are not often retracted. AI developers and big-tech companies often emulate unsafe drivers with insurance. Models rushed into deployment without adequate testing and safety guardrails often lead to bias, misinformation, and lack of interpretability (Bender et al., 2021). Black-box decision-making coerces a lack of accountability. ChatGPT has a preference for left-leaning viewpoints in 14 of the 15 different political orientation tests (Rozado, 2023). Motoki et al. (2024) find notable evidence in favor of ChatGPT having a political bias in favor of the Labour Party in the UK, Democrats in the US, and Lula in Brazil. In addition, Hazra & Serra-Garcia (2025) illustrates individuals have a limited ability to assess the accuracy of an LLM as a search engine but are often overconfident in their assessment abilities. Bad AI lie-detectors often hinder truth detection where individuals teamed with such AI models often perform at levels below their intrinsic capability (Bhattacharya et al., 2024).

At this juncture, we bring forward a prominent economic theory, *intertemporal consumption theory*, that offers insights into how people make trade-offs between present and future outcomes. Using this, we provide an analogy to evaluate the long-term impact of AI-assisted decisions.

**Intertemporal consumption theory.** Evaluating the stability of the economy at large, economic theory on intertemporal consumption theory examines how individuals consider factors such as income fluctuations, interest rates, and future expectations to allocate their earnings to consumption and savings. Developed by Ando & Modigliani (1963), the Life-Cycle Hypothesis (LCH) posits how individuals maintain a stable standard of living throughout their lifetime by saving (hence, consuming only a part of their income) during their working years and using the savings to maintain a similar standard of living as before during retirement. The Permanent Income Hypothesis (PIH), introduced by Friedman (1957), argues that consumption is determined

by an individual's expected permanent income rather than transitory changes in earnings. Both models posit that individuals partake in consumption smoothing, borrowing during low-income periods, and saving during high-income periods. Nonetheless, factors such as liquidity limits, income volatility, and behavioral anomalies frequently result in divergences from these theoretical forecasts.

Technical debt bears a larger societal cost of job displacement. Large-scale development and deployment of AI are expected to disrupt job stability, create wage polarization, and shift traditional career structures. The primary assumption of stable career trajectories in the LCH may no longer hold. AI-driven economies might produce multiple earnings peaks due to career shifts, mid-life retraining, and the rise of gig work, altering the traditional savings-consumption pattern. Under the PIH framework, AI-induced job insecurity may reduce the predictability of future earnings, leading individuals to base consumption decisions on their current income rather than long-term expectations. This shift would result in higher precautionary savings, particularly among workers whose occupations are at a greater risk of automation. Researchers found a 21% decrease in the weekly number of posts in automation-prone jobs (writing, app-, web development) compared to manual-intensive jobs after the introduction of ChatGPT[1].

> **P1:** Fueled by competition, the AI arms race piles up technical debt by destabilizing economies and making jobs insecure, a precursor to the disruption of traditional consumption smoothing.

### 2.2. Unchecked, Impractical Automation

Rapid uptake in AI-based automation (e.g., AI agents) is encouraging as well as alarming. Paslar (2023) identified a range of creative industries, including analysts, designers, and technicians, are at the frontier of getting impacted by AI automation. While Acemoglu & Restrepo (2019) argued that automation generally introduces new tasks where the existing labor force may have a competitive advantage, current trends in adopting automotive workflows are often rushed, if not impractical. We clearly observe the shift from 'assistance' to 'automation' in the context of the use of AI, which is so rapid and abrupt that it warrants further scrutiny in the lens of the future of work.

For example, Rony et al. (2024) records the anxiety and uncertainty in the nursing diaspora regarding the possibility of their job displacements due to AI. Healthcare professionals expressed concern about the future value of their lifelong investment in skill development and learning, which is critical

for their professional roles in society. Current AI adoption practices focus on improving trust in algorithmic agents and chatbots in critical healthcare scenarios. However, it fails to replace the empathetic care, foundational in the patient-provider relationship (Nash et al., 2023). Benchmarks such as (Sun et al., 2024) promote the efficacy of AI systems helping in critical decision-making in healthcare (e.g., reducing ED wait times). Arguably, it is cheaper to build AI-based assistive technologies, especially those that are just a tailored version of API-based LLM models, but they fail to address the issue of likely labor replacement of existing workers who are currently performing these decision-making tasks to their full potential. This exacerbates the problem even more and calls for an introspective review of the AI researchers powering through benchmarks that do not consider the future of work as part of AI safety. Finally, low accountability due to AI reliance significantly diminishes ethical oversight in healthcare (Morley et al., 2020).

Similarly, AI assistance is taking over software industries in many ways with the stated benefit of improved human productivity. Peng et al. (2023) demonstrated that programmers became 55.8% faster in developing code when assisted by an AI programmer. While this is encouraging for employers, it also impacts the future of work in the field of software engineering. Review by (Necula, 2023) is a cautionary telltale that highlights the need for the software engineering profession to adapt to the changing landscape in the wake of generative AI to remain relevant and effective in the future. The CEO of a major software engineering company recently revealed that they would not be hiring any software engineers going into 2025[2]. We argue that this is unprecedented, and such a strategy does not bode well for the software engineers. In similar efforts, OpenAI aims to automate most of the software industry by building powerful AI programmers, clearly indicating a shift from assistive systems to systems that can replace human labor.

Even though Hoffman, on a more optimistic note, claims a structural change in the future of work: "It's job transformation. Human jobs will be replaced—but will be replaced by other humans using AI"[3], the distribution of AI-driven productivity improvements raises additional concerns since it may favor high-skilled workers and capital owners disproportionately while leaving lower-skilled individuals with static or declining salaries. This might make liquidity restrictions worse and make it harder for some groups to use optimal consumption smoothing, connecting back to the intertemporal consumption theory from Section 2.1. In sum,

> **P2:** AI-driven automation accelerates skill dis-

---

[1] https://hbr.org/2024/11/research-how-gen-ai-is-already-impacting-the-labor-market

[2] https://fortune.com/2024/12/18/agentic-ai-salesforce-marc-benioff/

[3] https://www.cnn.com/2024/06/20/business/ai-jobs-workers-replacing/index.html

parity, favoring high-skilled workers while displacing lower-skilled labor, necessitating urgent workforce adaptation.

## 2.3. Declining Shared Prosperity

In society, power confers control. Acemoglu & Robinson (2013) argue for inclusive institutions that allow and encourage participation by the economic agents in economic activities as crucial for sustained economic growth and shared prosperity. Shared prosperity ensures an equitable distribution of economic pie, thus benefiting all segments of society and not just the powerful. The World Bank defines shared prosperity as the growth of income or consumption of the bottom 40% of a population (Narayan et al., 2013). In contrast, extractive institutions, which concentrate power and wealth in the hands of a few, hinder economic development.

Generative AI firms often operate as extractive institutions, where the benefits of its advancements are disproportionately concentrated among a select few rather than being widely distributed. Leaders in the AI paradigm have frequently emphasized the disruptive potential of these technologies, often taking an anti-worker stance - replacement of human labor (Landymore, 2023); some even resorting to careless rhetoric (Finance Yahoo, 2024): "Some creative jobs maybe will go away. But maybe they shouldn't have been there in the first place." As AI systems are trained incessantly to become increasingly capable of automating a broader range of tasks, the labor market experiences a surge in the supply of skills that were once considered valuable. The increased supply reduces the economic worth of skilled human labor, placing workers whose expertise overlaps with AI-driven automation at heightened risk of wage stagnation or job displacement (Restrepo, 2023), while investors and stakeholders remain largely unaffected with their wealth derived from capital ownership rather than wage labor. Concentrating economic power within a handful of dominant AI firms further exacerbates existing inequalities as it rent-seeks into the already-existing gap between capital owners (with business assets) and the employed labor force (workers). This sums into:

> **P3:** The extractive nature of generative AI reinforces structural disparities while diminishing worker bargaining power and economic security.

## 2.4. Uneven Democratization, when Looked Globally

Generative AI reeks of uneven democratization across the global market. Economic resources remain asymmetrical across regions, social classes, and industries. Proponents of AI research emphasize its potential to enhance productivity and expand knowledge generation. However, more often than not, benefits are concentrated among corpora-

tions, nations, and individuals with the resources to develop, implement, and regulate these technologies.

Restrictive access to resources takes a blow for a nation to be able to adopt and build AI systems for its citizens. Hazra & Serra-Garcia (2025) highlights uneven trust in AI systems in information-gathering tasks, showing that respondents from the US are most calibrated. This result naturally extends to the ability to perceive the impact of AI, which will vary dramatically depending on AI use and awareness. The uneven resources to build and investigate AI systems are evident from the AI Index Report[4], which shows the geographical clustering of AI research papers, patents, and conferences in high-income countries. Unfortunately, and critically, World Bank (2018) shows how insufficient digitization in sectors such as healthcare and agriculture in low-income countries can lead to gaps in the data required for training AI models. The uneven distribution of AI infrastructure—such as cloud computing, data centers, and high-quality training datasets—further entrenches technological dependency, where lower-income nations remain consumers rather than producers of AI innovations.

The global implications of this uneven democratization extend beyond labor markets to governance and policymaking. High-income countries, particularly those with dominant AI firms, dictate the norms and ethical guidelines surrounding AI development, while lower-income nations struggle to regulate and adapt these technologies to their socio-economic realities. This asymmetry reinforces technological and data colonialism (Couldry & Mejias, 2020), where the benefits of AI remain concentrated in the hands of a few, exacerbating global inequalities rather than alleviating them.

Uneven democratization underscores exploitation, hereby increasing rent-seeking and, thus, technical debt. In an open letter, artists criticized OpenAI for using them as "free bug testers, PR puppets," expressing concerns over the company's approach to integrating AI into creative fields[5]. Advanced regulatory frameworks such as EU's General Data Protection Regulation[6] enforce stringent data privacy protections. In contrast, lower-income countries lack institutional autonomy to implement data governance. In the Global South, data protection laws encompass more than just regulations; they also entail raising awareness of privacy and

---

[4] https://aiindex.stanford.edu/wp-content/uploads/2024/04/HAI_AI-Index-Report-2024.pdf

[5] https://www.washingtonpost.com/technology/2024/11/26/openai-sora-ai-video-model-artists-protest/

[6] https://eur-lex.europa.eu/eli/reg/2016/679/oj

data protection issues[7].

> **P4:** Uneven democratization leaves lower-income countries data-colonialised and dependent on external AI innovations.

## 2.5. Impaired Learning and Knowledge Creation

### 2.5.1. HARMS IN LEARNING

Generative AI has been heavily embraced by students, educators, and knowledge workers. The temptation to produce essays, homework, or tasks requiring critical thinking skills using AI, however, threatens the basic core of how humans learn and create knowledge. Over-reliance on AI for research, writing, or problem-solving bypasses the cognitive processes that are essential to a deep understanding of the subject matter. Recent work Bastani et al. (2024) shows how students attempt to use GPT-4 as a "crutch" during practice math problem sessions and, when successful, perform worse on their own. Without having students directly engage with the challenging material, generative AI can impair their critical thinking abilities.

Writing is one of the most important pillars of education, enabling learners to critically engage with the topics they study. In *The Rise of Writing* Brandt (2014) argues that the "information economy's insatiable demand for symbol manipulation—'knowledge work'—has forced many workers to reorient their labor around the production of prose" (Laquintano & Vee, 2024). The ways people write in modern workplaces and universities are closely connected and influence each other.

There is no rigid separation between these different types of writing— ideas, practices, and conventions flow between them and often overlap. Rampant use of generative AI in writing creates *algorithmic monoculture* (Kleinberg & Raghavan, 2021) that in turn decreases content as well as linguistic diversity (Padmakumar & He, 2023; Kobak et al., 2024). Such reliance also risks student to lose their own writing voice and individual expression. Leading public and private universities have evaluated college essays over 50 years for several qualities, such as the ability to show leadership and overcome hardships, creativity, and community service as part of their admissions decisions. The kinds of questions these institutions ask, the qualities they seek, and the responses they receive have changed many times and have been shaped by the cultural trends of our times. Using AI for college essays leads to homogenization at an unprecedented scale (Moon et al., 2024), sacrificing some of the authenticity and personal expression these essays were designed to offer (Thompson, 2022).

Generative AI has also significantly impacted scientific research. These range from researchers using AI for generating novel research ideas (Lu et al., 2024; Boiko et al., 2023; Si et al., 2024) as well as using them to evaluate scientific ideas (aka peer review using AI) (Lu et al., 2024). While the use of generative AI for scientific research offers benefits, these benefits might be skewed to only top scientists with considerable expertise in their respective areas. Recent work from Toner-Rodgers (2024) shows how top scientists posses *absorptive capacity* (Cohen et al., 1990) leveraging their domain knowledge to prioritize promising AI suggestions, while others waste significant resources testing false positives. These findings have broader implications for junior researchers, especially early-career PhD students. AI is affecting peer review. While far from perfect, the use of generative AI in peer review pushes low-quality scientific criticism at a massive scale. Recent work from Latona et al. (2024) shows how peer reviews assisted by artificial intelligence, in particular, LLMs, negatively influence the validity and fairness of the peer-review system, a cornerstone of modern science. Last but not least, generative AI is impacting behavioral and social science research that often relies on useful contributions from human participants in the form of surveys or task-specific data (Hämäläinen et al., 2023; Gilardi et al., 2023; Ziems et al., 2024; Argyle et al., 2023). Given how LLMs misportray and flatten identity groups (Wang et al., 2024a), they have the potential to bias or negatively influence findings in social science research.

### 2.5.2. THE RISKS OF AI-GENERATED SLOP: CALL FOR BETTER AI DETECTABILITY

While the future of generative AI is debatable, millions of people are battling with AI-related questions of their own, such as *Is my reviewer 2 actually ChatGPT? Is my student's essay AI-generated, or is it just repetitive, low effort, and full of clichés?* (Herrman, 2024). Garbled and obviously generated AI text is abundant but, by definition, easy to spot and dismiss. However, like any predictive model, AI detection tools can lead to false positives and should not solely determine disciplinary policy (Fowler, 2023). Additionally, while state-of-the-art AI detectors like Pangram Labs (Emi & Spero, 2024), GPTZero (GPTZero, 2023), and Binoculars (Hans et al.) are excellent at spotting zero-shot or few-shot AI-generated text, it becomes impossible to detect AI-generated text when it is fine-tuned.

OpenAI now allows fine-tuning GPT4-o on millions of tokens at less than 50$, making it feasible for individuals and small organizations to create customized language models that can effectively evade detection. This democratization of fine-tuning capabilities, while useful, also complicates the landscape of AI detection and raises important questions about the long-term viability of automated AI detection tools. Major AI companies have the capability to imple-

---

[7]https://carnegieendowment.org/posts/
2024/02/data-protection-regulation-in-the-
global-south?lang=en

ment robust watermarking systems in their models. If all companies adopted watermarking, then society (and even the industry long-term) might benefit from having clearer guardrails around AI-generated content.

However, any company that adds watermarking may worry about making their outputs more constrained or less "natural" than those of rivals. This could lead to lost market share or lower perceived performance compared to competitors. Each company thus has an incentive to *not* watermark in order to maximize its own product's appeal, resulting in an equilibrium where no one does—or at least no one does comprehensively. This situation—a misalignment between what is best for society or the industry collectively and what is best for each individual firm—closely mirrors the **Prisoner's Dilemma** (Axelrod & Hamilton, 1981; Poundstone, 2011) or more general **Collective Action Problems** (Olson, 2012; Ostrom, 1990) in economics.

> **P5:** In short, the rampant use of generative AI in education and research threatens the future of learning. This is further exacerbated by the difficulty of detecting AI-generated content.

### 2.6. Copyright Failing to Safeguard Human Labor

In the 1884 Supreme Court case Burrow-Giles Lithographic Co. v. Sarony (bur, 1884), the court addressed the definition of "author" within the context of copyright law. Justice Miller, delivering the unanimous opinion, referenced Worcester's dictionary, stating that an author is *he to whom anything owes its origin; originator; maker; one who completes a work of science or literature.* This definition is more important now than ever, especially as AI trained on years of human labor produces content with a polish that emulates—and may potentially oust—the works of creators (Ginsburg, 2025). Much of the success of generative AI depends on the high quality of human data available on the internet. With the rising popularity of generative AI, researchers have investigated the data used to train such models (Bandy & Vincent, 2021; Carlini et al., 2023; Dodge et al., 2021).

Several investigations have revealed that state of the art open open-weight/closed-form large language models are trained on copyright-protected data (Mishcon de Reya LLP). As a matter of fact, one of the largest players in the AI safety field, *Anthropic*, was sued by authors for copyright infringement for training its AI models on the Books3 dataset, which contains pirated ebooks (Peters, 2024). In one of the biggest generative AI-related copyright cases, OpenAI was sued by The New York Times for unpermitted use of Times articles to train large language models (Grynbaum & Mac, 2023). More recently, Meta's chief executive backed the use of the LibGen dataset, a vast online archive of books, despite

warnings within the company's AI executive team that it is a dataset "that is known to be pirated" [8].

For large-scale AI models, the raw volume of texts and the multitude of rights holders lead to very high licensing or clearance costs. A conventional Coasean approach would say that if there were zero transaction costs, the parties could bargain for a mutually beneficial outcome. But in reality, the transaction costs of contacting, negotiating with, and securing licenses from innumerable authors or creators are enormous. Because these costs are so high and because no single licensing clearinghouse exists that covers all written content, the friction to compliance is multiplied. When transaction costs are high enough, parties often choose noncompliance if they anticipate that paying a lawsuit settlement or even statutory damages (should they lose in court) will cost less than trying to bargain with millions of individual copyright owners (Coase, 2013; Landes & Posner, 2003).

Fair use is an affirmative defense that can be raised in response to claims by a copyright owner that a person is infringing a copyright. Generative AI firms repeatedly justify their use of copyrighted data as material for training models as fair use (Lemley & Casey, 2020; Henderson et al., 2023), given how the final model does not verbatim reproduce/regurgitate the content it was trained on. In 2023, The Atlantic published a story about the use of Books3 [9], a corpus of 191,000 pirated books, used to train LLMs. The same year, The Authors Guild conducted a large-scale survey with over 2,000 members. They found unanimous opposition among authors to their works being used without permission and compensation. The recent proposal made by ministers in the UK would allow LLM companies to train their AI systems on public works unless their owners actively opt-out. (The Guardian, 2024).

False promotions lead many to believe that the disruptive impact of "AI" on the creative economy implies that these systems possess genuine creative abilities. Such an axiomatic truth is further compounded by misleading evaluations (Porter & Machery, 2024; Alexander, 2024), reductionism, and general fear-mongering (Goetze, 2024; Hullman et al., 2023). Creative professionals now face a difficult choice: they must either demonstrate that their work has some unique, irreplaceable quality or find themselves forced to compete with AI on factors like cost and speed. Based on criticism from the community, recently publishing gamut Harper Collins has tied up with "a large tech company" to offer authors a one-time 2500$ non-negotiable fee to include their books as a part of the training data. We believe these practices are harmful because such an amount does not qual-

---

[8] https://regmedia.co.uk/2025/01/10/pacer_kadrey_vs_meta_1.pdf

[9] https://huggingface.co/datasets/defunct-datasets/the_pile_books3

ify as fair compensation. And while AI might not verbatim generate copyrighted data because of guardrails, by training on a creator's entire lifetime worth of work, these models mimic their style, unique voice, and creative expression, thereby risking their livelihoods (Porquet et al., 2024).

> **P6:** The kind of exploitation that big generative AI firms do by training on copyright-protected data in the garb of fair use causes unforeseen risk to musicians, writers, artists, and other creatives. The lack of transparency, licensing deals, and fair payment further devalues their labor.

## 3. Our Recommendations

Generative AI differs from previous technological changes due to its ability to automate complex cognitive tasks across several sectors of the economy. The manner in which foundation models are built is shaped by the expectation and desire to reach Artificial General Intelligence (a.k.a. AGI). While it is unclear what the end goal of AGI is, a lot of its success lies in automation. It is no surprise that automation leads to job displacement and lower wages and is thus inconsistent with shared prosperity.

One way to deal with the consequences could be simply slowing progress. While we find logic in this approach, there are arguments in the literature that suggest that such efforts may not be a sustainable and highlight the differential impacts of suppressing technological progress (Acemoglu & Johnson, 2023; Bostrom, 2014; Acemoglu & Robinson, 2012; Mokyr, 1992a). Efforts to slow down progress often shift it's locus and may not be effective globally, as other regions or entities might continue development. This may lead to an uneven and uncontrolled advancement landscape. Additionally, such interventions are often entangled with broader geopolitical motives. In the light of all these arguments we argue for more proactive measures, such as pro-worker strategies where we allow society to adapt to the change and the makers of AI models to simultaneously adapt to the interests of society, possibly with measures such as collective licensing, fair compensation, etc.

> **R1:** Given AI's potential to cause job displacement, we require policy measures to support affected workers. It is crucial for governments to expand and modernize unemployment insurance, establish comprehensive social safety nets, and offer retraining programs for vulnerable workers. An AI researcher must be aware of such consequences and address safeguards when developing AI-based automated systems.

The transformative potential of AI is more likely to benefit workers if development is not concentrated among a few giant corporations. Historically, major technological breakthroughs tend to come from new companies rather than established ones, especially when the incumbents are as powerful as the giants. Recently, DeepSeek (Liu et al., 2024), a relatively new international player, has displaced the top-performing AI, o1 (from OpenAI).

> **R2:** Promoting worker interests in AI development aligns naturally with efforts to increase competition and reduce the dominance of big tech companies over the industry's direction. AI researchers must focus on open AI that includes open data, open weights, and transparency through the training and adaptation pipelines.

Back in 2023, OpenAI released their AI detector, which they claimed as unreliable and recommended against use[10]. Since then, the community has significantly improved AI detection with firms like Pangram (Emi & Spero, 2024) matching expert performance (Russell et al., 2025). There are existing issues with AI detection, especially false positives, but that should not discount the fact that there is a dire need for reliable detectors. Fraud detection, security screening, lie detection, and mammograms all have some degree of false positives, but it can be argued that society has benefited from these technologies to a certain extent.

> **R3:** From an AI consumer's perspective, there needs to be accountability on whether something is solely written by AI, written entirely by humans, or written with AI assistance. Educational institutions, academic conferences, and universities particularly need to educate themselves on the state of the art for AI detection and update their policies accordingly.

While sophisticated adversaries can bypass watermarking techniques (Jovanović et al., 2024; Saberi et al., 2023), that should not prevent policymakers from imposing this. More funding can encourage research in this area.

> **R4:** Content from generative AI poses a huge threat to democracy and public opinion, and there needs to be a policy that mandates all generative AI companies to watermark their content. AI makers must put safeguards in place to protect against this and actively measure the effect of guarding through improved AI safety benchmarks.

Generative AI companies seldom share information about their training corpus due to legal risks, especially those

---

[10]https://openai.com/index/new-ai-classifier-for-indicating-ai-written-text/

around copyright violations. In the past, AI firms have threatened to quit a whole continent when pressured to reveal their training data (Reuters, 2023). While tools to trace, filter, and automate data licensing lack the necessary scale and effectiveness to meet current demands, it is crucial to have government intervention that forces the development of such tools. Recent work from Wang et al. (2024b) proposes an economical solution to address copyright challenges in generative AI. Inspired by cooperative game theory and the Shapley value, they introduce a framework called Shapley Royalty Share (SRS) that fairly compensates copyright owners based on their contributions to AI-generated content.

> **R5:** To safeguard creative labor and artists policy, policymakers should mandate disclosure of training data, build tools to audit large-scale pre-training, and introduce royalty-based incentives that compensate them based on their contribution. Until then, creators should restrict AI companies from exploiting them by training on their work for very little compensation (Broussard, 2024).

Letting large AI firms such as OpenAI, Anthropic, and Google significantly influence AI policy can be viewed as a form of regulatory capture (Levine & Forrence, 1990), a concept drawn from public choice theory and Stigler's theory of economic regulation (Stigler, 2021). Regulatory capture is the process by which industries (or other special interest groups) exert undue influence on the agencies or lawmakers meant to regulate them. Researchers can independently assess the potential benefits and risks of AI applications without being guided by market incentives. Organizations representing consumers, privacy advocates, and workers can highlight concerns that might otherwise be overlooked (Korinek & Vipra, 2025). Including smaller firms and startups in discussions can help ensure that regulations do not artificially shut out new entrants.

More specifically, for workers, traditional unions like the Writers Guild of America have taken the lead in negotiating contracts through collective bargaining that address the impact of AI in creative industries, such as scriptwriting. Labor unions are increasingly pushing for robust regulatory frameworks to govern AI deployment in workplaces. Their advocacy ensures that workers' rights, privacy, and job security are considered in national and international AI policies. Workers can build alliances with advocacy groups. New initiatives, such as the Campaign to Organize Digital Employees by the Communications Workers of America, aim to unionize tech and video game workers.

On the other hand, industry coalitions such as the Global Partnership on Artificial Intelligence are platforms where AI makers can be involved in human-centered lobbying that protects creators' rights and values instead of lobbying to

weaken policy. Working closely with organizations like Responsible AI can provide appropriate training, assessments, and toolkits to help AI makers strengthen governance, enhance transparency, and scale innovation responsibly. Stakeholders and AI Makers need to have equal representation and voice in decision-making such that corporate lobbying does not win over worker rights.

> **R6:** To avoid the pitfalls of regulatory capture, both workers and AI makers must take a role in lobbying and advocating policies to protect the future of work.

## 4. Alternative Views

### 4.1. Market Forces Will Naturally Adapt

This position paper focuses on protecting labor markets from disruption caused by generative AI, which may understate the market's natural capability to attune and evolve. Smith (2002) argued that competitive markets and the division of labor respond flexibly to changing conditions, including technological innovations. Autor (2015) shared a similar view, stating how automation complements labor, raises output in ways that lead to higher demand for labor, and interacts with adjustments in labor supply. We have historical precedence in the Industrial Revolution, which, despite displacing artisanal workers, led to the emergence of totally new industries and job categories (Mokyr, 1992b). Similarly, the digital revolution did the same in terms of creating new employment sectors in software, digital content, and online services (Autor et al., 2003).

Furthermore, Aghion (1990) argue that innovation-driven economic growth depends on the incentives for firms to put money into new technologies and processes; excessive constraints or labor market rigidities can lower these incentives and slow growth that would raise overall welfare. Generative AI may follow a similar trend to other revolutions in the past, and hence, government intervention, when not needed, can restrict economic freedoms and affect innovation, ultimately hindering the creation of new jobs (Friedman, 2016). This may suggest that the paper's concerns about permanent labor displacement may be overemphasized.

### 4.2. AI Safety Should Only Focus on Catastrophic Risks

Hendrycks et al. (2023) provides an extensive overview of the potential catastrophic AI risks (or existential risks a.k.a. x-risks) into fours categories: malicious use of AI by bad actors, deployment of unsafe AIs to win AI race, complex nature of human interactions with AI systems, and rogue AI. A popular argument for AI safety community has been to exclusively focus on x-risks as the stake is highest compared to other effects such as future of work.

However, Kasirzadeh (2025) argues that the discourse on

x-risks from AI mainly focuses on decisive risks (e.g., rogue AI takeover) but misses its counterpart: accumulative x-risks (e.g., systemic erosion of economic structures by misuse of AI). In fact, our position directly connects to the latter. Declining shared prosperity, impaired learning, and uneven democratization of AI resources can fuel polarization and destabilizing effects (retrospectively analogous to the industrial revolution), which may seem short-term but are likely to have cascading dynamics leading to catastrophic transitions.

Secondly, human frailty in assessing the long-term catastrophic risks stemming from AI can be biased and inadequately founded (Swoboda et al., 2025); hence, it cannot be the only aspect of AI safety. We take a balanced position, highlighting the additional need to focus on the future of work for AI safety.

### 4.3. Technical Solutions Over Policy Interventions

This view contends that technical approaches to AI development provide a more effective path forward than policy-focused labor market interventions. Researchers can focus on building techniques that allow for meaningful human control. Through critical oversight of evolving AI systems, they can ensure that humans remain central to the decision-making process across industries. Policy interventions around labor markets, while having good intentions, could be premature, quickly outdated, or even counterproductive, given the rapidly evolving nature of AI. Instead generative AI firms should take the responsibility of developing AI that shares the goal of putting humans first.

## 5. Related Work

A limited number of recent works began addressing the long-term socio-economic impacts of generative AI through theory-backed arguments and empirics. Eloundou et al. (2023) showed impacts of LLMs exhibiting traits of general-purpose technologies—around 80% of the U.S. workforce could have at least 10% of their work tasks affected by the introduction of LLMs, while approximately 19% of workers may see at least 50% of their tasks impacted. Felten et al. (2023) presented a methodology to systematically assess the extent of such impact—the top occupations exposed to LLMs include telemarketers, post-secondary teachers, and legal services. Interestingly, they also showed a positive correlation between wages earned and exposure to AI.

Korinek (2024) discussed the potential labor market impacts and the rate at which they unfold, bringing novel challenges for policymakers. Hui et al. (2024) found that generative AI models like ChatGPT and DALLE2 have negatively impacted freelancers, even the most high-performing ones on online platforms. Additionally, (Howard, 2019) discusses

that AI tools can boost productivity but may negatively impact workers' well-being through surveillance and automated management of their workloads. While most prior work focused on measuring and quantifying AI's impact on labor markets and worker productivity, this position paper uniquely argues for making the future of work a core consideration in AI safety research, providing concrete recommendations for protecting worker interests through technical safeguards, policy interventions, and broader stakeholder engagement in AI development—aspects that have received limited attention in existing AI safety literature.

## 6. Conclusion

We argue that the current AI safety paradigm poses a restrictive view of the long-term consequences of AI. Just focusing on technical concerns remains unhinged to critical socio-economical consequences such as economic justice and labor market stability. We posit that safeguarding human labor should be taken under the wings of AI safety as rapid, unchecked AI-based automation continues to erode human agency, creative labor, and incentives to learn. Only by prioritizing the future of work through a pro-worker governance framework can AI remain a tool for upholding shared prosperity rather than a highway for labor displacement.

## Impact Statement

We highlight critical concerns about the lack of focus on the future of work in the AI safety discourse. We put forward a series of arguments with scholarly evidence to discuss several medium and long-term risks of generative AI on meaningful labor. Our recommendations include strategies that can correct the current course of AI safety research to protect labor rights.

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
