# OpenReview forum: "Position: AI Safety should prioritize the Future of Work"
_ICML.cc/2025/Position_Paper_Track — ICML 2025 Position Paper Track oral_

### Official Review · Reviewer_8NWJ · 2025-03-05

**Significance:** 4
**Argument Clarity:** 3
**Rating:** 5
**Confidence:** 5

**Questions:**

1. is there any evidence that measures like watermarking, AI accountability or data transparency will have positive impact on the labour market?

**Discussion Potential:**

4

**Paper Summary:**

The paper begins with a high-level overview of the impact of AI (ML) systems on society, and work in particular. An analysis of key risks related to how AI impacts work follows, including technical debt, unchecked automation, declining shared prosperity, uneven democratization of AI at global scale, impaired knowledge creation, and failure of copyright law. Authors employ several economic theories to support their analysis. The analysis broadens the traditional scope of risks considered in AI safety research. This is followed by several recommendations that focus on social security, promoting competition through openness, data transparency, accountability, and watermarking. The paper ends with a presentation of alternative positions.

**Position:**

Yes

**Position In Title:**

Yes

**Related Work:**

4

**Strengths And Weaknesses:**

Strengths:
- the position is well defined, and strongly supported both with reasoning, available evidence and theoretical works (mainly in economic theory)
- the exploration of risks related to the future of human work is based on a critical analysis of the dominant perspective on AI safety, and serves to broaden it
- the recommendations connect, in interesting ways, concerns over the future of work with measures related to intellectual property, and also demonstrate how well understood proposals on data transparency or watermarking, usually seen as accountability measures, could be part of policy focused on work-related issues

Weaknesses:
- in the abstract, authors state that they "argue in favor of a robust international copyright anatomy supported by implementing collective licensing that ensures fair compensation mechanisms for using data to train AI models". Yet the paper does not include any such proposal.
- I would instead highlight what the authors describe in the abstract as a "pro-worker framework", and if possible expand the analysis how recommendations related to AI accountability can also deal with labour challenges from automation.
- the scope of the position paper is very broad, which I appreciate. but it runs the risk that some of the issues are covered too briefly - seems that the authors tried to include a very large body of ideas and theories into the piece. for example, the piece on AI slop offers a very narrow analysis, and then only in passing mentions game theory concepts.

**Support:**

4

---

> ### Author Rebuttal · Authors · 2025-04-01
>
> Thank you for taking the time to review our submission! We are pleased that you found our position well-defined and strongly supported through critical analysis and rightly addressed through interesting recommendations.
>
> > __W1: Paper does not include a proposal for international copyright anatomy__
>
> Thank you for pointing this out. You are right; we advocate for a robust international copyright framework and collective licensing mechanisms in Section 2.6 and in R3. However, since this is a position piece, we focus on guiding principles rather than presenting a detailed policy proposal. Developing such a proposal would require global structures for licensing bodies—topics we believe need in-depth policy treatments, currently beyond the scope of this paper.
>
> > __W2: Expand how recommendations related to AI accountability can also deal with labour challenges from automation__
>
> This is a great suggestion! Our recommendation on AI accountability rests on these points: 1) an accountable AI research must understand the (creative) workers' rights and decide where the benefit of automation (e.g., in solving a scientific discovery problem) outweighs the cost of inadvertently affecting a worker's life's worth effort. 2) accountability should percolate to the global governance where the right policies that allow the natural progression of technological development while protecting the workers' rights should be implemented.
>
> > __W3: AI slop offers a very narrow analysis, passing mentions game theory concepts__
>
> We acknowledge that the broad scope limited the depth of coverage in certain areas. Our brief reference to game theory in the context of AI slop intends to illustrate the structural incentives that discourage practices like watermarking. We used concepts like the Prisoner’s Dilemma to frame industry behavior as a collective action problem. Complete expansion of this theory can help us provide systemic solutions to AI content quality and accountability, which we leave as a future work.
>
> > __Q1: Evidence that measures like watermarking, AI accountability or data transparency will have positive impact on the labour market?__
>
> We do not find strong evidence yet that measures like watermarking, AI accountability, or data transparency have already had positive impacts on the labor market. However, we can draw prospective conclusions from analogous situations, such as data transparency initiatives enhanced collective auditing by workers [1] and biased predictions primarily caused by biased training data [2]. However, we largely think this stems from the public's lack of collective understanding of the various elements of watermarking, accountability, and transparency. [3] The AI makers implementing these measures need to explain the fundamentals to the general public, who, in turn, will gradually get accustomed to them. The situation is synonymous with ‘nutrition labels’. People take time to recognize, utilize, and get accustomed to before making decisive actions [3].
>
> [1] Demystifying technology for policymaking: Exploring the rideshare context and data initiative opportunities to advance tech policymaking efforts. arXiv, 2024.
> [2] Biased programmers? Or biased data? A field experiment in operationalizing AI ethics. ACM Conf on Economics and Computation, 2020.
> [3] https://innovating.news/article/watermarks-are-just-one-of-many-tools-needed-for-effective-use-of-ai-in-news/
>
> _We will update our next version with these additional points._

---

### Official Review · Reviewer_q8hG · 2025-03-09

**Significance:** 4
**Argument Clarity:** 2
**Rating:** 4
**Confidence:** 4

**Questions:**

- To what extent are different claims in the current paper conditional on certain AI capabilities thresholds? For instance, does it matter that certain systems can be "shown" to beat humans in some knowledge work benchmark? Or, in other cases, does it not matter is AI can actually produce "good" educational content (because it will be used either way?)
- Which outcomes in the paper have tension with each? (For instance, will the production of AI-generated slop actually give back some degree of leverage to humans who produce non-slop?)
- How will global concerns interact with attempts to empower worker leverage?

**Discussion Potential:**

3

**Paper Summary:**

This paper argues that the AI Safety field (broadly construed, i.e. inclusive of industry labs, academics, and more) should explicitly focus on issues related to economic inequality, job substitution, and disruption to labor markets. The work draws on theories around rent-seeking and consumer economics to highlight several specific pathways to harms (beyond e.g., general wide-scale job loss). The paper describes 6 specific risks (P1-P6) and 6 specific recommendations (R1-R6). Briefly, these map to: job insecurity, skill disparity, bargaining power for labour, global power disparities (mirroring colonial power relations), harms to learning, and copyright concerns.

**Position:**

Yes

**Position In Title:**

Yes

**Related Work:**

2

**Strengths And Weaknesses:**

Strengths:
- Extremely clear position, and it is also clear how this might translate directly into AI safety research organizations and project.
- Some helpful coverage of theories from outside ML to draw on (collective action problems, intertemporal consumption, rent-seeking, monoculture). Could have impact on new projects that draw from across these lenses.

Weaknesses:
- Some variance in how clear each of the recommendations are (R1 was clear; R2 seemed very broad, and it wasn't clear that open AI will promote worker interests at all; R3 was very clear; R4 and R5 has some overlap and didn't seem as clearly related to the high level position about the AI safety field and it's focus).
- More generally, the paper starts with a very clear, focused position, but then becomes very scattered between sections as a number of theories are brought in (which, as noted above, is also partially a strength).
- Most importantly, the current draft has some tension around the actual level of AI capabilities that are reached. Specifically, there are concerns both around the idea that AI can substitute jobs (thereby massively reducing labor leverage, and so on) but also that AI will create technical debt, work poorly, etc. Of course it's possible that indeed AI capabilities reach some ceiling, and are still deployed widely and create economic harms (without much benefit) but it would be helpful from the perspective of the position here to take a stance on this question, or acknowledge the tension.

- Very Minor: purely from a reader cognitive load perspective, I was expecting Px and Rx to match up (e.g., was confused about P4 referencing colonial power relations and R4 referencing AI watermarking).

**Support:**

2

---

> ### Author Rebuttal · Authors · 2025-04-01
>
> Thank you for your review! We are pleased that you regard our position as extremely clear and influential for AI safety research, and our theoretical coverage can impact future work.
>
> > __W1: Variance in recommendations (R2 is broad, R4 and R5 have some overlap)__
>
> R2 highlights how open AI systems can promote worker interests by fostering competition and enabling participation from smaller companies and worker cooperatives. Workers can better advocate for their interests when they understand how AI systems make decisions that affect employment, workflows, and labor conditions. Currently, a lack of transparency—such as OpenAI and Anthropic withholding pre-training data—limits this understanding.
>
> R4 and R5 approach generative AI content from different societal angles—R4 focuses on the consumer perspective, while R5 emphasizes policy initiatives from AI producers. Despite their distinct viewpoints, they overlap significantly, as consumers and producers remain interconnected within society.
>
> > __W2: Paper is initially clear, but later becomes scattered with economic theories (also partially a strength)__
>
> This is a great comment. While we intentionally introduced various economic theories to enrich our discussion, we consciously avoided extensive theoretical detail to maintain relevance and accessibility to the ML community. We tried to provide sufficient theoretical context so that interested researchers can leverage them to develop further relevant claims and insights.
>
> > __W3: Tension around the actual level of AI capabilities__
>
> The current tension around AI stems from uncertainty about its true capabilities. While some fear it could replace human labor, others point to its limitations and inefficiencies. However, the key issue is that even the perception of powerful AI can harm workers. Exaggerated narratives can gaslight employers to cut hiring, reduce training, or restructure jobs prematurely, weakening labor power and degrading job quality—regardless of AI’s actual performance. For instance, recent AI-generated images in Studio Ghibli’s art style went viral. Despite lacking the intentionality and depth of human-made animation, it fueled the hype around AI’s creative potential. Such hype distorts expectations and accelerates premature adoption. In a similar feat, a sci-fi magazine halted submissions after being overwhelmed by a surge of AI-generated stories. AI makers and reviewers must carefully manage AI narratives alongside technical advances.
>
> > __W4: (minor) P4 does not match with R4__
>
> Sorry! We will reorganize them in camera-ready.
>
> > __Q1: Which claims conditional on AI capabilities thresholds__
>
> We iterate our answer for W3 that most of our claims hold irrespective of how capable AI is. Public perception of AI is largely driven by hype, not the actual capability. This is evident by viral AI-generated images mimicking Studio Ghibli’s art style, which fueled hype around AI’s creative potential despite lacking the intentionality and depth of human creativity.
>
> > __Q2: Which outcomes in the paper have tension with each?__
>
> 1. Our position underscores that AI-generated content creates "algorithmic monoculture" that reduces diversity while flooding markets with mediocre content. Paradoxically, this could potentially create new value for genuine human creativity that stands out from AI-generated content.
> 2. P2 notes how automation favors high-skilled labor and displaces lower-skilled labor. But P5 and P6 note that even high-skill creative and scientific domains are getting saturated with AI-generated noise unless workers develop "absorptive capacity", a term we use to describe one’s ability to leverage AI effectively without becoming obsolete.
> 3. The current copyright system is too slow and fractured to protect creatives but enforcing it strictly may slow AI progress or raise access costs.
>
> > __Q3: How will global concerns interact with attempts to empower worker leverage?__
>
> Industry coalitions such as the [Partnerships on AI](https://en.wikipedia.org/wiki/Partnership_on_AI) or [Global Partnership on Artificial Intelligence](https://en.wikipedia.org/wiki/Global_Partnership_on_Artificial_Intelligence) are global platforms where AI makers can be involved in human-centered lobbying that protects creators' rights and values instead of lobbying to weaken policy. Organizations like Responsible AI can provide appropriate training, assessments, and toolkits to help AI makers strengthen global governance, enhance transparency, and scale innovation responsibly. Equal representation of workers and AI Makers in decision-making can ensure corporate lobbying does not win over worker rights. Additionally, labor unions are increasingly pushing for robust regulatory frameworks to govern AI deployment in workplaces. Their advocacy ensures that workers’ rights, privacy, and job security are considered in national and international AI policies.
>
> _We will add these points in camera-ready._

---

> > ### Comment · Reviewer_q8hG · 2025-04-05
> >
> > My apologies, I posted my initial response as an "Official Comment". In short, I appreciate the comments and remain in favour of "Accept".

---

> > > ### Author Response · Authors · 2025-04-06
> > >
> > > Thank you for your reply and voting for acceptance.

---

### Official Review · Reviewer_qPBz · 2025-03-14

**Significance:** 4
**Argument Clarity:** 4
**Rating:** 5
**Confidence:** 5

**Questions:**

How would you address the AI safety community that focuses exclusively on catastrophic risk? This seems like the most substantive alternative view to this work, but is not really addressed in this paper.

**Discussion Potential:**

3

**Paper Summary:**

This paper argues that considerations of AI safety should prioritize the future of work. This paper discusses the systemic risks that AI poses to labor. These include the pace of development, uneven distribution of resources, and so on. This paper argues for several recommendations to respond to these risks — for example, by bolstering international copyright law, pushing for fair compensation for using training data.

**Position:**

Yes

**Position In Title:**

Yes

**Related Work:**

3

**Strengths And Weaknesses:**

**Strengths**

* This paper adds a much-needed perspective to AI safety discourse. The labor side of the AI debate is talked about far too little, and it is certainly not given the platform it deserves. This paper and perspective is somewhat novel, and it is well argued.
* This is one of those papers that I want to cite already - I need it to be published!!!
* This paper is well written. It ties economic theories (like intertemporal consumption theory) nicely to the points it is trying to make.
* The point about the extractive nature of genAI, and about the disproportionate economic impacts, is especially well made.
* The structure of the paper, with its highlighted points, is nice.

**Weaknesses**

* The discussion of this work could be further strengthened. How do we preserve the fun and creative sides of human jobs? How do we ensure that human-AI teams outperform AI or human teams alone? How do we maintain social connection at work? Etc
* The abstract does not properly summarize the contributions of this paper. It focuses only on copyright, but the recommendations in the main text include job displacement efforts, watermarking, etc.
* Some of the recommendations are a bit loose and could be made more concrete and actionable. For example, R6. How should workers and AI makers engage in lobbying and advocacy?
* The alternative views section of this paper is quite weak. It is also missing the alternative view that I am most aware of: that AI safety must focus on catastrophic risk above all other concerns, because if p(doom) > 0 then this is infinitely more important than any other issue. So we can forget about workers! 4.2 is especially vague, and contains no citations.

## Update after rebuttal
Like the other reviewers have said: clear accept.

**Support:**

4

---

> ### Author Rebuttal · Authors · 2025-04-01
>
> Thank you for your review. We appreciate that you acknowledge our position, which adds a much-needed perspective on AI safety, is clearly written, and should be published!
>
> > __W1: How do we preserve the fun and creative sides of human jobs?__
>
> Thank you for bringing this up! We believe one of the effective ways to preserve this is by advocating for fairer contracts that explicitly protect human agency. Recent examples, such as the negotiations by the [Writers Guild of America](https://www.wga.org/contracts/know-your-rights/artificial-intelligence) and [SAG-AFTRA](https://www.sagaftra.org/sag-aftra-strikes-video-games-over-ai), demonstrate how collective action can set boundaries around AI use in creative industries. We stand a better chance of safeguarding the human elements of work if more organizations can be institutionalized to adopt policies aimed at fairer contracts.
>
> To ensure human-AI collaboration is genuinely productive, people must understand both the strengths and weaknesses of humans and AI independently. Current discourse in AI is often misguided in the hype, which makes humans believe that human-AI teams can often do the heavy lifting, leading to unfavorable outcomes. E.g., AI may struggle with planning in writing but often excels at revising [1]—so teaching students to use AI for revision, not idea generation, can lead to better outcomes.
>
> [1] When combinations of humans and AI are useful: A systematic review and meta-analysis. Nature Human Behavior, 2024.
>
> > __W2: The abstract does not include job displacement efforts, watermarking__
>
> For brevity, we clubbed watermarking, copyright, and similar topics under "copyright anatomy supported by implementing collective licensing" and job displacement efforts as recommendations for a "pro-worker framework of global AI governance."
>
> > __W3: How should workers and AI makers engage in lobbying and advocacy?__
>
> For workers:
>
> 1. Traditional unions like the Writers Guild of America have taken the lead in negotiating contracts through collective bargaining that address the impact of AI in creative industries, such as scriptwriting.
> 2. Labor unions are increasingly pushing for robust regulatory frameworks to govern AI deployment in workplaces. Their advocacy ensures that workers’ rights, privacy, and job security are considered in national and international AI policies.
> 3. Workers can build alliances with advocacy groups. New initiatives, such as the Campaign to Organize Digital Employees by the Communications Workers of America, aim to unionize tech and video game workers.
>
> For AI Makers:
>
> 1. Industry coalitions such as the Global Partnership on Artificial Intelligence are platforms where AI makers can be involved in human-centered lobbying that protects creators' rights and values instead of lobbying to weaken policy. Working closely with organizations like Responsible AI can provide appropriate training, assessments, and toolkits to help AI makers strengthen governance, enhance transparency, and scale innovation responsibly.
> 2. Stakeholders and AI Makers need to have equal representation and voice in decision-making such that corporate lobbying does not win over worker rights.
>
> > __W4a: Missing the alternative view that AI safety must focus on catastrophic risk above all other concerns__
>
> Great point! We feel adding this alternative view will strengthen our paper.
>
> > __W4b: Missing citations in Sec 4.2__
>
> We add citations that highlight the necessity of human-centric technical innovation.
>
> [1] Position: data-driven discovery with large generative models. ICML, 2024
> [2] When combinations of humans and AI are useful: A systematic review and meta-analysis. Nature Human Behavior, 2024
>
> > __Q1: Address the AI safety community that focuses exclusively on catastrophic risk.__
>
> First, our position does not advocate abandoning efforts to contain possible existential (x)-risks. The discourse on x-risks from AI mainly focuses on decisive risks (rogue AI takeover) but misses its counterpart: accumulative x-risks (e.g., systemic erosion of economic structures by misuse of AI) [1]. In fact, our position directly connects to the latter. Declining shared prosperity, impaired learning, and uneven democratization of AI resources can fuel polarization and destabilizing effects (retrospectively analogous to the industrial revolution), which may seem short-term but are likely to have cascading dynamics leading to catastrophic transitions.
>
> Secondly, human frailty in assessing the long-term catastrophic risks stemming from AI can be biased and inadequately founded [2]; hence, it cannot be the only aspect of AI safety. We take a balanced position, highlighting the additional need to focus on the future of work for AI safety.
>
> [1] Two types of AI existential risk: decisive and accumulative. arXiv, 2024
> [2] Examining Popular Arguments Against AI Existential Risk: A Philosophical Analysis. arXiv, 2025
>
> _We will update our next version with these additional points._

---

> > ### Comment · Reviewer_qPBz · 2025-04-02
> >
> > Thank you for the thorough response!

---

> > > ### Author Response · Authors · 2025-04-02
> > >
> > > Thank you again for your response and encouraging review!

---

### Official Review · Reviewer_Whdq · 2025-03-24

**Significance:** 4
**Argument Clarity:** 4
**Rating:** 5
**Confidence:** 4

**Questions:**

See above.

**Discussion Potential:**

4

**Paper Summary:**

This paper argues that AI safety concerns should prioritize the needs of workers, particularly those who may see their livelihoods significantly impacted by generative AI tools. The authors argue that the speed of development, coupled with a lack of democratization of the technology, engenders outcomes that diminish education and impair the ability of workers to be fairly compensated for their labor. They anticipate that this may have markedly different outcomes relative to previous technological revolutions, wherein new industries and sectors were created, offsetting job loss to those whom the technology replaced. The authors then make several recommendations, including the need for government intervention, open-source training, and increased adoption of AI water-marking schemes.

## Update after rebuttal

Clear accept, I'd give it a spotlight or equivalent. Have gone back to reread this since reviewing it just for my own benefit -- exactly what a position paper should be.

**Position:**

Yes

**Position In Title:**

Yes

**Related Work:**

4

**Strengths And Weaknesses:**

* I thought this was a fantastic position piece. The position was clearly stated and defended with evidence. The topic is timely, and over the course of eight pages, the authors lay out what I find to be a convincing argument. It is certainly true that the progress in AI will threaten the future of work in many industries. None of us know the extent of future job displacement, but it's reasonable to believe that we are in an age that rivals the transformative change seen only a few times in the course of modern history.
* I thought the recommendations were sensible, although I have a couple of thoughts about two of them.
    1. Regarding the need for preemptive measures to put unemployment measures in place, it seems that an alternative route could be to curb the pace of development. Rather than assuming that it's only a matter of time before certain jobs are gone, one could imagine interventions that cause the pace of progress to slow. While perhaps undesirable to researchers, it's clear that measures like export controls on GPUs can slow progress (although it's worth acknowledging that this is generally associated with oftetimes unrelated political posturing).
    2. Regarding the open-source training recommendation, I'm not sure if this is in the best interest of AI safety. If data, training pipelines, and models are all open-sourced, it creates a situation in which bad actors can use these technologies in the most harmful possible ways. At present, my understanding is that frontier labs offer fine-tuning APIs that automatically check to see whether the data being fine-tuned on is malicious or harmful (although, I haven't used these APIs personally, so this could be wrong). If this is the case, it would stand to reason that it's relatively difficult to fine-tune, say, an evil version of o1. However, if the weights for o1 were released, it would be incredibly easy for *anyone* to train this kind of model. And given that such models are generally much more capable across a broad spectrum of tasks (not just math, coding, and reasoning a la DeepSeek), this would represent an avenue for widespread harm.
* Neither of these points are criticism; they're just reactions that I had to the paper. But again, I thought this paper was great, hit exactly the right note, is important to publish at a conference like ICML to increase awareness of these issues, and, lastly, was very well written.

**Support:**

4

---

> ### Author Rebuttal · Authors · 2025-04-01
>
> Thank you so much for taking the time to review our submission! We are pleased to see your encouraging points about our position being clear, well-written, and defended with sufficient evidence, warranting a timely publication at ICML.
>
> > __C1: It seems that an alternative route could be to curb the pace of (AI) development__
>
> It is indeed an interesting consideration. While we find logic in this approach, there are arguments in the literature that suggest that slowing progress may not be a sustainable approach and highlight the differential impacts of suppressing technological progress [1, 2, 3, 4]. Efforts to slow progress down often shift the locus of progress and may not be effective globally, as other regions or entities might continue development. This may lead to an uneven and uncontrolled advancement landscape. Additionally, as you rightly mentioned, these interventions are often entangled with broader geopolitical motives. Instead, we argue for more proactive measures, such as pro-worker strategies where we allow society to adapt to the change and the makers of AI models to simultaneously adapt to the interests of society, possibly with measures such as collective licensing, fair compensation, etc.
>
> [1] Power and Progress: Our Thousand-Year Struggle Over Technology and Prosperity. Public Affairs.
> [2] Superintelligence: Paths, Dangers, Strategies. Oxford University Press.
> [3] Why nations fail: The origins of power, prosperity, and poverty. Crown Currency.
> [4] The Lever of Riches: Technological Creativity and Economic Progress. Oxford University Press.
>
> >  __C2: Not sure if open sourcing is in the best interest of AI safety__
>
> We agree with your concern that open-sourcing powerful AI models can lower the barrier to misuse, especially by bad actors. However, closed-source models are not protected from bad actors either. Jailbreak is increasingly common, even with close-sourced models. However, the lack of transparency restricts good actors from auditing, aligning, and reproducing such undesired behaviors stemming from progressive research that can make powerful models less susceptible to attacks. Further, the open-source ecosystem is critical to prevent monopoly, which should enforce close-soured models to embrace AI safety research more seriously than achieving organizational standards. In addition to that, we believe a possible middle ground should be structured access: models can be shared for research under strict licensing, sandboxed environments, or with red-teaming partnerships.
>
> _We will update our next version with these additional points._

---

### Decision · Program_Chairs · 2025-04-30

**Decision:**

Accept (oral)

**Comment:**

The paper has unanimous, strongly positive feedback, and should be accepted to the conference.